# Spt5 C-terminal repeat domain phosphorylation and length negatively regulate heterochromatin through distinct mechanisms

**Sarah MacKinnon, Viviane Pagé, Jennifer J. Chen, Ali Shariat-Panahi, Ryan D. Martin, Terence E. Hébert, Jason C. Tanny***

Department of Pharmacology and Therapeutics, McGill University, Montreal, Canada

\* jason.tanny@mcgill.ca

**Data Availability Statement:** All relevant data are within the manuscript and its Supporting Information files.

## Abstract

Heterochromatin is a condensed chromatin structure that represses transcription of repetitive DNA elements and developmental genes, and is required for genome stability. Paradoxically, transcription of heterochromatic sequences is required for establishment of heterochromatin in diverse eukaryotic species. As such, components of the transcriptional machinery can play important roles in establishing heterochromatin. How these factors coordinate with heterochromatin proteins at nascent heterochromatic transcripts remains poorly understood. In the model eukaryote *Schizosaccharomyces pombe (S. pombe)*, heterochromatin nucleation can be coupled to processing of nascent transcripts by the RNA interference (RNAi) pathway, or to other post-transcriptional mechanisms that are RNAi-independent. Here we show that the RNA polymerase II processivity factor Spt5 negatively regulates heterochromatin in *S. pombe* through its C-terminal domain (CTD). The Spt5 CTD is analogous to the CTD of the RNA polymerase II large subunit, and is comprised of multiple repeats of an amino acid motif that is phosphorylated by Cdk9. We provide evidence that genetic ablation of Spt5 CTD phosphorylation results in aberrant RNAi-dependent nucleation of heterochromatin at an ectopic location, as well as inappropriate spread of heterochromatin proximal to centromeres. In contrast, truncation of Spt5 CTD repeat number enhanced RNAi-independent heterochromatin formation and bypassed the requirement for RNAi. We relate these phenotypes to the known Spt5 CTD-binding factor Prf1/Rtf1. This separation of function argues that Spt5 CTD phosphorylation and CTD length restrict heterochromatin through unique mechanisms. More broadly, our findings argue that length and phosphorylation of the Spt5 CTD repeat array have distinct regulatory effects on transcription.

## Author summary

Formation of transcriptionally silent heterochromatin involves deposition of repressive histone modifications such as methylated histone H3 lysine 9 (H3K9me), which creates a

**Funding:** This work was funded by the Canadian Institutes of Health Research (https://cihr-irsc.gc.ca/)(MOP-130362 and PJT-173356 to J.C.T.) and the Natural Sciences and Engineering Research Council (https://www.nserc-crsng.gc.ca/)(RGPIN 03661-15 and RGPIN-2020-05174 to J.C.T.). The funders had no role in study design, data collection and analysis, decision to publish, or preparation of the manuscript.

**Competing interests:** The authors have declared that no competing interests exist.

compact, inaccessible chromatin state. Paradoxically, deposition of H3K9me can involve transcription at the target locus. How the molecular machinery that forms heterochromatin interfaces with the transcription apparatus remains unclear. In the fission yeast *S. pombe*, H3K9me deposition is coupled to engagement of nascent heterochromatic transcripts by the RNA interference (RNAi) pathway, as well as RNAi-independent mechanisms. We show that heterochromatin formation in this system is negatively regulated by the ancient elongation factor Spt5 through its C-terminal repeat domain (CTD). We demonstrate that phosphorylation of the CTD by Cdk9 restricts inappropriate action of the RNAi pathway at euchromatic locations, whereas the length of the repeat array restricts RNAi-independent heterochromatin formation. Thus, different alleles of the same gene impact heterochromatin through distinct mechanisms. Our findings suggest that Spt5 is a key modulator of transcription-coupled heterochromatin establishment and have implications for the role of the CTD in transcriptional regulation.

## Introduction

Transcriptional silencing of repetitive DNA elements proximal to centromeres and telomeres, as well as many developmentally regulated genes, occurs through formation of compacted chromatin structures termed heterochromatin. These structures are propagated through cell division and are thought to underlie epigenetic inheritance [1]. Defects in the establishment or propagation of heterochromatin are associated with genome instability and cancer [2–4]. Histone modification is central to the formation and propagation of heterochromatin. Methylation of histone H3 on lysine 9 (in pericentric regions) or on lysine 27 (at developmental genes) creates a binding site for chromodomain proteins that mediate chromatin compaction, likely through mechanisms involving liquid-liquid phase separation [5–7].

How heterochromatin is established and maintained at a particular location in the genome is a question that remains unresolved. Evidence from multiple eukaryotic model systems indicates that nascent RNA transcripts produced from heterochromatic loci help to direct heterochromatin formation [8–10]. This has been characterized in detail in the model eukaryote *Schizosaccharomyces pombe* (*S. pombe*), in which heterochromatin nucleation is linked to processing of nascent transcripts by the RNA interference (RNAi) pathway [11–15]. In *S. pombe*, heterochromatin nucleation sites, often corresponding to non-coding, inverted repeat sequences within heterochromatin domains, are transcribed by RNA polymerase II (RNAPII) and processed by the double-strand-RNA specific RNase Dicer (Dcr1) and RNA-directed RNA polymerase (Rdp1) into siRNAs [16,17]. The siRNAs are subsequently bound by the Argonaute family protein Ago1 of the RNA-induced transcriptional silencing complex (RITS) [18–20]. siRNA-bound RITS engages heterochromatic nascent transcripts through base-pairing interactions [21]. RITS also recruits the histone methyltransferase Clr4, which tri-methylates histone H3 at lysine 9 (H3K9me3) [22]. At pericentromeric heterochromatin outer repeats, transcription of the inverted *dg/dh* repeats occurs during S phase, providing a temporal window for RNAi-dependent heterochromatin establishment [23,24]. RITS is retained at heterochromatin nucleation sites because its Chp1 subunit is a chromodomain protein that binds to H3K9me3. This creates a self-reinforcing loop that sustains heterochromatin at defined genomic locations [8,13].

H3K9me3 formed through this RNAi-linked pathway is also bound by other chromodomain-containing effector proteins that carry out chromatin condensation, notably the HP1 ortholog Swi6 [25]. In addition, Clr4 itself contains a chromodomain that recognizes

H3K9me3. This "read-write" mechanism maintains chromatin localization of Clr4, and also leads to methylation of the adjacent nucleosomes and heterochromatin spreading in *cis* [26]. The extent to which pericentric heterochromatin spreads is restricted by multiple mechanisms: boundary elements, action of Epe1 (a demethylase-like protein that removes H3K9me3), and action of other euchromatic chromatin-modifying factors that promote histone turnover [27–29]. The siRNAs produced from heterochromatic loci are also prevented from spreading in *trans*, i.e. targeting a homologous sequence distant from their site of origin [30,31]. Restriction of *trans*-silencing seems to involve regulation of transcription elongation and proper mRNA 3' end processing, as mutants that disrupt these processes have been found to allow siRNA-directed heterochromatin formation at ectopic locations [30,32,33]. This highlights the fact that the RNAi machinery must act within the context of the general RNA polymerase II (RNA-PII) transcriptional apparatus to establish heterochromatin.

Although the RNAi-dependent pathway is required for *S. pombe* pericentromeric hetero-chromatin formation, parallel mechanisms that are independent of RNAi also contribute, and may play more important roles at other heterochromatic loci. These include post-transcriptional silencing mechanisms involving transcription termination factors, the nuclear exosome complex, and mRNA elimination factors. Transcriptional silencing mechanisms involving stress-responsive transcription factors or a histone deacetylase complex have also been implicated [34–40]. Interestingly, contribution of RNAi-independent mechanisms to pericentric heterochromatin can be revealed in RNAi mutants, which can be suppressed by inactivation of several factors regulating RNAPII elongation or mRNA processing [41,42]. This further underscores the complex interplay between transcription and heterochromatin formation.

The *S. pombe* model system has been powerful in revealing mechanisms of co-transcriptional assembly of heterochromatin. However, many key questions remain, including how the general transcription machinery interfaces with heterochromatin proteins and what prevents these interactions at transcribed genes. We investigated these questions through study of Spt5, a conserved, essential RNAPII elongation factor. Eukaryotic Spt5 is required for processive RNAPII elongation, and also promotes RNAPII stability and enhancer function [43–46]. Cryo-electron microscopy structures of Spt5 in complex with elongating RNAPII show that Spt5 maintains the RNAPII transcription bubble by forming "clamps" around the DNA template and the exiting RNA. The clamps are formed by NGN (NusG N-terminal) and KOW (Kyprides, Ouzounis, Woese) domains that are conserved in prokaryotic, eukaryotic, and archaeal Spt5 orthologs [47,48]. Eukaryotic Spt5 orthologs also contain a C-terminal domain (CTD; also called a C-terminal region or CTR) that is comprised of multiple repeats of a 6–9 amino acid motif [49,50]. Like the analogous CTD on the RNAPII large subunit, the Spt5 CTD interacts with factors involved in elongation, mRNA processing, and histone modification. Whereas the Rpb1 CTD is phosphorylated by multiple kinases, the Spt5 CTD is primarily targeted by Cdk9 [51]. Cdk9-dependent phosphorylation of the Spt5 CTD enhances RNAPII elongation rate and processivity, and promotes co-transcriptional histone modifications [45,52–55]. These effects are mediated at least in part by Rtf1 (Prf1 in *S. pombe*), the only known phospho-specific CTD interactor [56,57]. Rtf1 directly stimulates the ubiquitin-conjugating activity of Rad6, leading to histone H2B monoubiquitylation (H2Bub1) and subsequent H3 lysine 4 methylation (H3K4me) [58]. It also stimulates elongation rate by binding directly to RNAPII [59,60]. Rtf1 requires the Polymerase Associated Factor (PAF) complex for stable association with elongating RNAPII; Rtf1 and PAF have been found to have largely (but not completely) overlapping functions [57,60,61]. The unphosphorylated CTD binds to mRNA capping and 3'-end processing factors [49,62,63]. The Spt5 CTD is generally more variable in repeat sequence and number between species than the RNAPII CTD. *S. pombe* Spt5 harbors

an unusually regular and uniform CTD comprised of 18 repeats of a nonapeptide motif (TPAWNSGSK) that is phosphorylated by Cdk9 on Thr1 [64].

Prf1/Rtf1 and PAF complex subunits were identified as key factors restricting RNAi-dependent formation of heterochromatin at an ectopic location in *trans*. We thus hypothesized that the Spt5 CTD also acts to limit co-transcriptional heterochromatin formation. We tested this in Spt5 CTD mutants that differed in the sequence of individual CTD repeats and/or the number of repeats in the CTD array. Our results indicated that repeat number in the CTD and phosphorylation status of the repeats distinctly regulate RNAi-dependent and RNAi-independent heterochromatin pathways.

## Results

### Spt5 CTD phosphorylation blocks trans-silencing by heterochromatin-derived siRNAs

To determine if the Spt5 CTD has a shared function with Prf1/Rtf1 and the PAF complex in negatively regulating siRNA-mediated *de novo* heterochromatin formation, we used a dual *ade6+* reporter system in which the *ade6+* marker is present at its endogenous, euchromatic location and at a second location within pericentric heterochromatin [33]. The heterochromatic *ade6+* acts as a source of *ade6+*-derived siRNAs (Fig 1A). In wild-type cells, these siRNAs

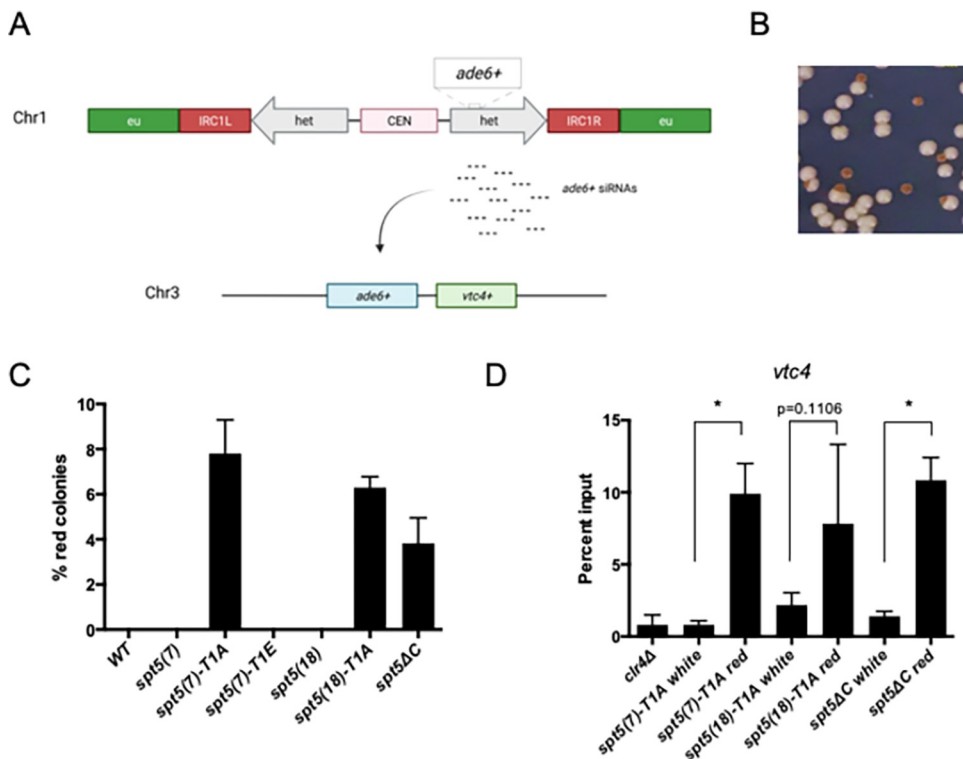

**Fig 1. *spt5-T1A* and *spt5ΔC* allow siRNA-mediated heterochromatin formation in *trans*. (A)** Schematic of reporter gene system used to detect siRNA-directed *trans*-silencing. The copy of *ade6+* inserted into the heterochromatic *dg* repeats acts as an siRNA source. **(B)** Example image of *spt5(7)-T1A* colonies on adenine-limiting media. **(C)** Frequency of red colonies formed by *spt5* mutants and controls expressing *ade6+* from the *dg* locus. ~1000 cells from each genotype were plated on adenine-limiting media. Error bars indicate SEM (n = 4). **(D)** Anti-H3K9me3 ChIP-qPCR using cells originating from red or white colonies as indicated; *clr4Δ* was included as a negative control. Percent input at *vtc4+* was normalized to *S. cerevisiae* spike-in. Error bars indicate SEM. Asterisks indicate significant differences for the indicated comparisons [p<0.05, one-way ANOVA followed by Holm-Sidak's multiple comparison test (unpaired) with a single pooled variance; n = 3].

can only act on the heterochromatic *ade6+*, such that the euchromatic copy is fully expressed and all the cells form white colonies on adenine-limiting media. Mutations affecting PAF components (or other regulators of transcription elongation and mRNA export) allow the siRNAs produced from the heterochromatic *ade6+* to act in *trans*, resulting in *de novo* heterochromatin formation at the endogenous *ade6+* locus in a fraction of the mutant cells. These events are scored as red colonies on adenine-limiting media [33].

We introduced this reporter system into two sets of *spt5* mutant strains: one that harbored substitutions in T1 of the nonapeptide CTD motif in each of the 18 repeats in the full-length protein [*spt5(18)*], and one that harbored the same substitutions in the context of a truncated, 7-repeat CTD [*spt5(7)*] (see S1 Table) [65]. This allowed independent assessment of the effects of repeat length and T1 phosphorylation state in the assay. We also included a strain in which the entire CTD was deleted (*spt5ΔC*) [65]. The *spt5(7)-T1A*, *spt5(18)-T1A*, and *spt5ΔC* strains exhibited ~4–8% red colonies when plated on adenine-limiting media, a frequency similar to that observed for highly penetrant drivers of siRNA-mediated *trans*-silencing in this and similar reporter systems [31,32]. (Fig 1B and 1C). In contrast, wild-type controls, or *spt5-T1E* mutant strains, did not give rise to any red colonies. Thus, Spt5 T1 phosphorylation (pSpt5), or installation of a constitutive negative charge as a phosphomimetic at this position, blocks siRNA-mediated silencing in trans.

To confirm that the observed *trans*-silencing correlated with *de novo* heterochromatin formation, we first assessed the mitotic stability of the red colony phenotype in re-plating assays. As observed for other mutants that enhance trans-silencing, red colonies gave rise to a mixture of red and white colonies upon re-plating to adenine-limiting media, with red predominating over white. This is consistent with the red colony phenotype arising due to *de novo* heterochromatin formation (S1A Fig). We next performed anti-H3K9me3 ChIP-qPCR using cells derived from red and white *spt5-T1A* and *spt5ΔC* colonies. A *clr4Δ* strain, in which H3K9 methylation is absent, was included as a control for antibody specificity. H3K9me3 enrichment at *vtc4+*, a gene adjacent to the endogenous copy of *ade6+*, was increased between 4-fold and 12-fold in red colonies compared to white colonies for *spt5(7)-T1A*, *spt5(18)-T1A*, and *spt5ΔC* (Fig 1D) [33]. These changes were statistically significant in the *spt5(7)-T1A* and *spt5ΔC* strains. H3K9me3 enrichment at *vtc4+* in the red *spt5(18)-T1A* isolates was more variable but still suggestive of ectopic heterochromatin formation. Overall, these results indicated that pSpt5, independent of Spt5 CTD repeat number, is involved in repressing ectopic siRNA-mediated heterochromatin formation.

## Spt5 CTD phosphorylation prevents expansion of pericentric heterochromatin in *cis*

We next considered whether pSpt5 is important for regulating heterochromatin at its normal chromosomal locations. To this end, we utilized a *ura4+*-based reporter assay to detect heterochromatin spreading outside a boundary element flanking pericentric heterochromatin on chromosome I (*irc1*). The *irc1L::ura4+* reporter lies just outside this boundary and is fully expressed in wild-type cells, conferring sensitivity to the *ura4+* counterselection drug 5-fluoroorotic acid (5'FOA)(Fig 2A) [66]. *spt5-T1A* and *spt5ΔC* strains harboring this reporter showed enhanced growth compared to control in the presence of 5'FOA, suggesting that the reporter is silenced in some fraction of the mutant cells (Fig 2B and 2C). This was dependent on the positioning of *ura4+* outside *irc1L*, as all Spt5 CTD variant strains expressing *ura4+* from its endogenous locus grew similarly to controls (S2A Fig). These results suggest that *spt5-T1A* and *spt5ΔC* variants that allow siRNA-mediated ectopic heterochromatin nucleation also result in increased heterochromatin spreading at *irc1*.

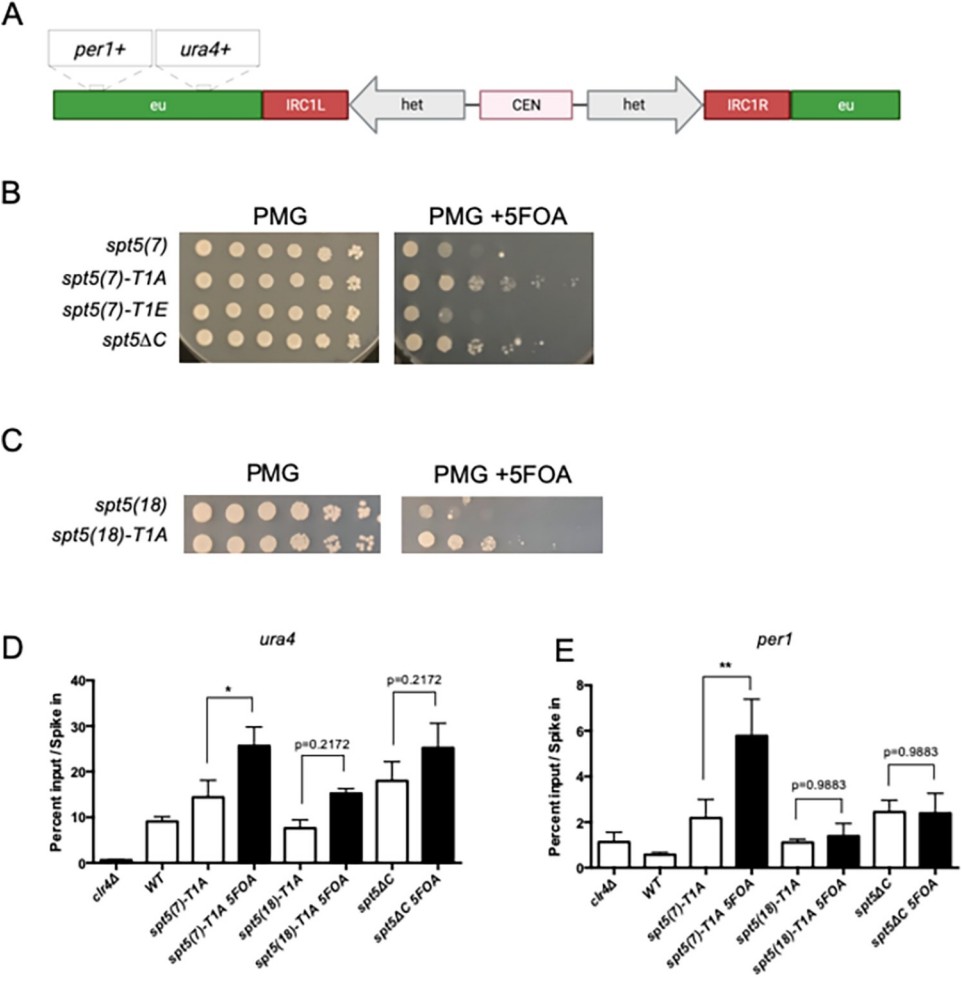

**Fig 2. *spt5-T1A* and *spt5ΔC* allow aberrant heterochromatin spreading at *irc1*. (A)** Schematic showing *ura4*⁺ reporter gene construct used to detect heterochromatin spreading at *irc1*. **(B) and (C)** Spot tests of indicated *spt5* mutants on control PMG media, PMG lacking uracil, and PMG with 5'FOA. Plates were incubated at 30 degrees for 7 days before imaging. **(D)** Anti-H3K9me3 ChIP-qPCR using cells of the indicated genotypes grown in non-selective media (white bars) or 5'FOA-containing media (black bars); *clr4Δ* was included as a negative control. Percent input at *irc1L::ura4*⁺ [27] was normalized to *S. cerevisiae* spike-in. Error bars indicate SEM. Asterisks indicate significant differences for the indicated comparisons [p<0.05, one-way ANOVA followed by Holm-Sidak's multiple comparison test (unpaired) with a single pooled variance; n = 4]. **(E)** As in (D) with a primer pair at *per1*⁺, located ~3kb distal to the centromere from *ura4*⁺.

To confirm that the increased growth on 5'FOA correlated with spread of heterochromatin, we performed anti-H3K9me ChIP-qPCR on the Spt5 CTD variant strains. We were unable to detect increases in H3K9me2 or H3K9me3 proximal to *irc1* in *spt5* mutants grown in non-selective conditions (S2B–S2D Fig). To test for the occurrence of aberrant heterochromatin spreading in a small percentage of the mutant cell population, we performed anti-H3K9me3 ChIP on isolates of the *spt5-T1A irc1L::ura4*⁺ and *spt5ΔC irc1L::ura4*⁺ strains grown under 5'FOA selection. We compared enrichment of H3K9me3 at *irc1L::ura4*⁺ to that for isolates grown in non-selective media. We observed a ~50% increase in *ura4*⁺ H3K9me3 enrichment in *spt5(7)-T1A irc1L::ura4*⁺ cells grown under 5'FOA selection compared to cells grown without selection. 5'FOA selection did not affect *ura4*⁺ H3K9me3 enrichment for *spt5(18)-T1A irc1L::ura4*⁺ or *spt5ΔC irc1L::ura4*⁺ strains (Fig 2D). ChIP analysis with primers located at

*per1*+, ~3 kb distal to *irc1L* with respect to the centromere, also revealed a significant increase in H3K9me3 enrichment in 5'FOA-grown *spt5(7)-T1A irc1L::ura4*+ cells compared to the same strain grown in non-selective media (Fig 2E). In contrast, we observed no such difference for *spt5(18)-T1A irc1L::ura4*+ or *spt5ΔC irc1L::ura4*+ strains (Fig 2E). We thus conclude that Spt5 CTD mutants compromise the *irc1L* heterochromatin boundary in a manner that is modulated by both the phosphorylation state and the length of the CTD repeat array, and that both length truncation and phospho-ablation are needed to trigger aberrant spreading of H3K9me3-marked heterochromatin.

## Truncation of the Spt5 CTD renders pericentric heterochromatin RNAi-independent

Our results pointed to a negative regulatory role for Spt5 CTD phosphorylation in RNAi-dependent heterochromatin formation. To test whether the Spt5 CTD may also negatively regulate RNAi-independent heterochromatin formation, we generated double mutants combining each of the Spt5 CTD variants with *ago1Δ*. The *ago1*+ gene encodes the sole Argonaute family protein in *S. pombe*, which mediates siRNA action on heterochromatin. Pericentric heterochromatin is strongly dependent on *ago1*+, but some residual, RNAi-independent heterochromatin is present in *ago1Δ* and can be enhanced in some elongation/mRNA export mutants [41,67]. We utilized the microtubule-destabilizing agent thiabendazole (TBZ) as a proxy for pericentric heterochromatin integrity; mutants deficient in pericentric heterochromatin impair centromere function and spindle-kinetochore attachment, resulting in sensitivity to TBZ. Strikingly, truncating the Spt5 CTD to 7 repeats rescued the TBZ-sensitivity of *ago1Δ* (Fig 3A). This phenotype was independent of pSpt5, as it was caused by *spt5(7)*, *spt5(7)-T1A*, and *spt5(7)-T1E* mutations (albeit more weakly in the T1 mutants). Moreover, complete removal of the Spt5 CTD by the *spt5ΔC* mutation was *not* able to rescue *ago1Δ* in this assay and displayed TBZ sensitivity on its own. This suggests that this mutation impacts additional mechanisms related to centromere function. An identical pattern of suppression was obtained when these Spt5 CTD mutations were combined with a knockout of the *S. pombe* Dicer ortholog, *dcr1Δ* (S3A Fig). These results suggested that reduction of Spt5 CTD repeat number may bypass the requirement for the RNAi pathway in formation of pericentric heterochromatin.

To further test this idea, we performed anti-H3K9me3 ChIP-qPCR with primers that amplify the *dh* repeat region within pericentric heterochromatin. Deletion of *ago1*+ in an otherwise wild-type background reduced H3K9me3 enrichment by ~6-fold. H3K9me3 enrichment was rescued to near-wild-type levels in the *ago1Δ* strain when combined with *spt5(7)* (Fig 3B). Thus, the requirement for *ago1*+ for pericentric H3K9me3 was bypassed by the *spt5 (7)* allele, but not *spt5(7)-T1A*, *spt5(7)-T1E*, or *spt5ΔC*. We made a similar finding using a primer pair located near the *irc1R* heterochromatin domain boundary, although the suppressive effect of *spt5(7)* did not reach statistical significance at this location (S3B Fig). These results confirm that reduced Spt5 CTD repeat number bypasses the requirement of *ago1*+ for pericentric heterochromatin and indicate that bypass is modulated by CTD T1 mutations.

We also measured the levels of heterochromatic transcripts in these strains using strand-specific RT-qPCR. Consistent with previous results, *ago1Δ* increased the levels of the transcript corresponding to the forward strand of the *dh* repeat (Fig 3C) [11]. This effect was abrogated in the *spt5(7) ago1Δ* double mutant. The *spt5(7)-T1E* and *spt5ΔC* variants failed to suppress, whereas the *spt5(7)-T1A* variant conferred weak suppression. The induction of the *dh* reverse strand transcript by *ago1Δ* was not statistically significant due to high variance between experimental repeats, but the fact that expression was increased at least 5-fold compared to wild-type was consistent with previous results (Fig 3C) [11]. In contrast, expression in the *spt5(7) ago1Δ*

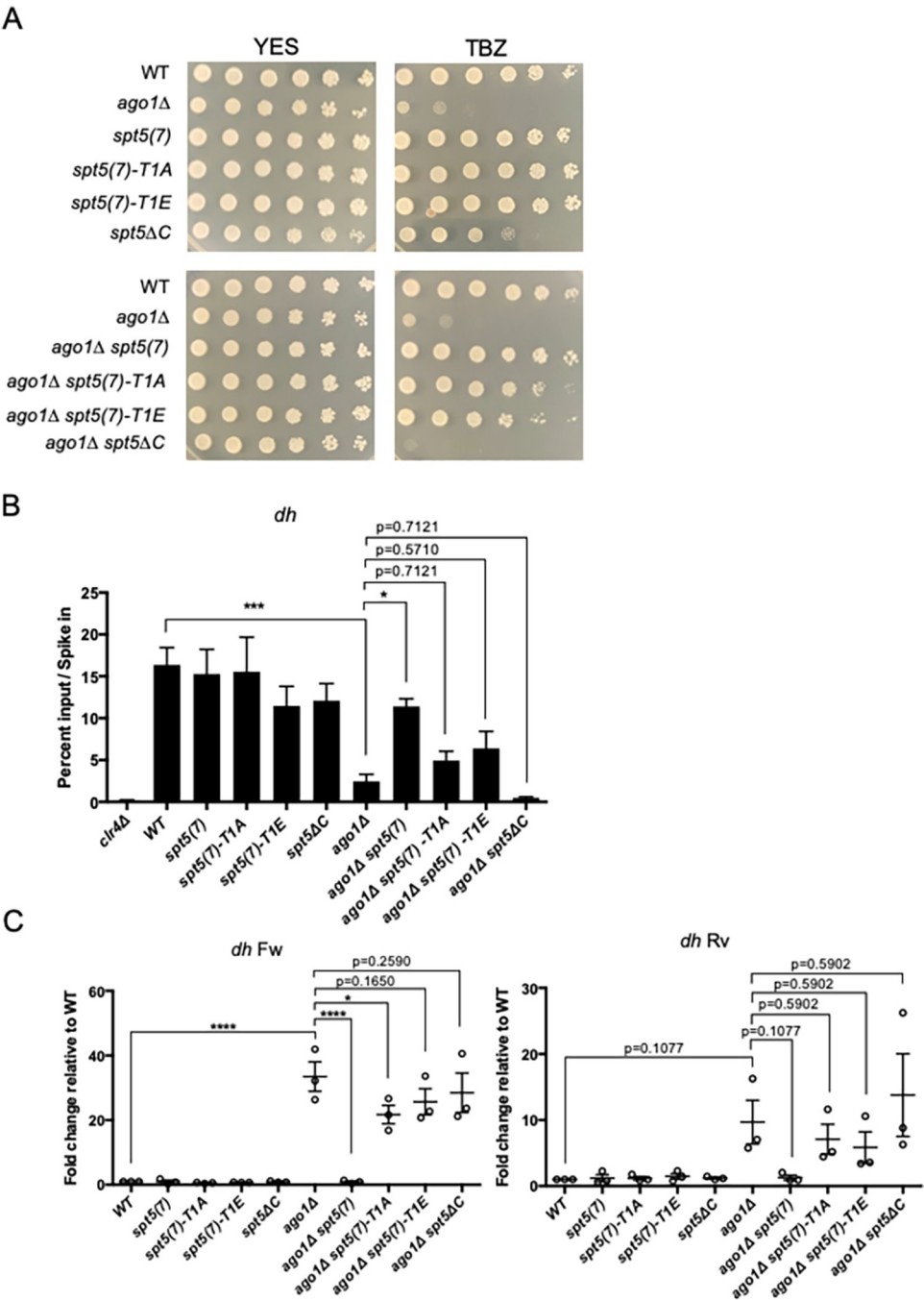

**Fig 3. A truncated Spt5 CTD allows pericentric heterochromatin formation in the absence of the RNAi pathway.** (**A**) Spot tests of the indicated strains on control media (YES) or YES containing thiabendazole (TBZ). Plates were incubated for 3 days at 30 degrees before imaging. (**B**) Anti-H3K9me3 ChIP-qPCR quantified with primers amplifying the *dh* repeat in the indicated strains. Percent input was normalized to *S. cerevisiae* spike-in. Error bars indicate SEM. Asterisks indicate significant differences for the indicated comparisons [$p < 0.05$ for 1 star, $p < 0.01$ for 2 stars, $p < 0.001$ for 3 stars, one-way ANOVA followed by Holm-Sidak's multiple comparison test (unpaired) with a single pooled variance; n = 4]. (**C**) Strand-specific RT-qPCR analysis of *dh* repeat transcripts in the indicated strains. Transcript levels measured with *dh* primers were normalized to *act1⁺* and the wild-type (WT) normalized value was set to 1 [$p < 0.05$ for 1 star, $p < 0.01$ for 2 stars, $p < 0.001$ for 3 stars, one-way ANOVA followed by Holm-Sidak's multiple comparison test (unpaired) with a single pooled variance; n = 3).

double mutant was identical to wild-type, indicating complete suppression of *ago1Δ* by CTD truncation. The double mutants expressing *spt5(7)-T1A*, *spt5(7)-T1E*, or *spt5ΔC* exhibited the same variable induction of expression as *ago1Δ* alone, again arguing that the complete *ago1Δ* suppression caused by a truncated Spt5 CTD requires the T1 residue.

To further probe the relationship between Spt5 CTD phosphorylation and repeat number, we constructed double mutants of full length Spt5 CTD T1 point mutants with *ago1Δ* [*ago1Δ spt5(18)-T1A* and *ago1Δ spt5(18)-T1E)*]. Growth of these strains on TBZ media showed that blocking or mimicking phosphorylation at this site did not rescue the sensitivity of *ago1Δ* (Fig 4A). Correspondingly, H3K9me3 enrichment levels in each double mutant are indistinguishable from enrichment levels in *ago1Δ*. (Figs 4B and S3C). Thus, pSpt5 restricts the RNAi-dependent heterochromatin pathway and heterochromatin spreading, whereas Spt5 CTD repeat number restricts RNAi-independent heterochromatin formation in a manner that depends on Spt5 CTD T1 (Fig 4C).

## Evidence that Prf1/Rtf1 is an effector for the Spt5 CTD in regulating heterochromatin

Given that pSpt5 creates a binding site for the Plus3 domain of Prf1/Rtf1, we queried *prf1* mutants for similar effects on heterochromatin regulation. Wild-type or mutant versions of Prf1 fused to a TAP-tag were expressed in the dual *ade6+* reporter strain [68]; the wild-type *prf1-TAP* did not give rise to any red colonies (Fig 5A). Point mutations in the Plus3 domain that impair Prf1 association with chromatin activated *trans*-silencing of euchromatic *ade6+*. This was true for *prf1-R227A*, known to abrogate phospho-specific binding to the Spt5 CTD, as well as *prf1-R262E* and *prf1-R296E*, which affect an as-yet unknown Plus3 domain interaction [68]. C-terminal truncation of Prf1 (*prf1ΔC1-345*) also allowed *trans*-silencing in this strain. This form of Prf1 is recruited to chromatin normally but lacks protein segments that bind to PAF complex components and to RNAPII [60,68,69]. A shorter C-terminal truncation of *prf1+* was previously shown to have the same effect [30]. These data are consistent with the notion that pSpt5 negatively regulates *de novo*, siRNA-dependent heterochromatin formation by promoting recruitment of functional Prf1/Rtf1 to chromatin.

The *prf1-R227A*, *prf1-R262E*, and *prf1ΔC1-345* mutations exhibited increased growth in the presence of 5'FOA in strains harboring the *irc1L::ura4+* reporter construct, whereas the *prf1-R296E* mutation, which had a weak phenotype in the dual *ade6+* reporter strain (and causes only partial loss of Prf1 chromatin binding), had no effect (S4A Fig). However, 5'FOA resistance in the *prf1-R227A irc1L::ura4+* strain was not associated with increased H3K9me3 at *irc1L::ura4+* by ChIP-qPCR (S4B and S4C). These data are similar to our findings for the *spt5(18)-T1A* and *spt5ΔC* mutants, suggesting that Plus3 domain mutations have effects on heterochromatin spreading that align with loss of CTD phosphorylation, rather than with CTD length truncation.

We next assessed the ability of *prf1* mutants to suppress *ago1Δ* effects on pericentric heterochromatin. We performed anti-H3K9me3 ChIP-qPCR to detect enrichment at the *dh* repeat in *prf1* and *prf1 ago1Δ* mutants. The average fold decrease caused by *ago1Δ* in the *prf1-TAP* background (~5-fold) was similar to that observed in previous experiments, although the decrease was not statistically significant due to variability in the signal in the *prf1-TAP* strain (Fig 5B). There were no significant differences in H3K9me3 enrichment at *dh* between *ago1Δ prf1-TAP* and *ago1Δ prf1R262E-TAP*, *ago1Δ prf1R227A-TAP*, or *ago1Δ prf1ΔC-TAP* (Fig 5B). Thus, *prf1* mutations do not suppress the effect of *ago1Δ* on H3K9me3 levels at pericentric heterochromatin. Strand-specific RT-qPCR analysis of *dh* repeat transcripts showed that the ~30-fold de-repression of the forward strand transcript caused by *ago1Δ* was maintained in

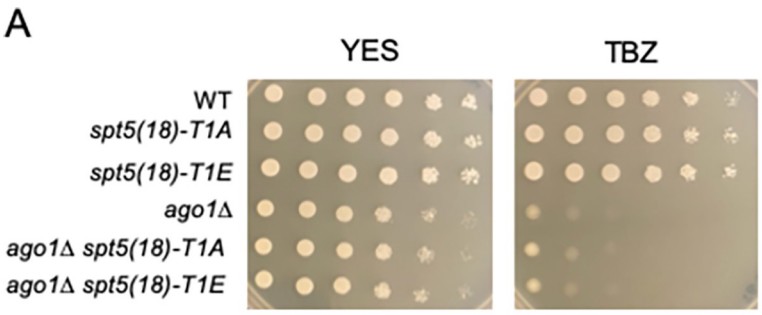

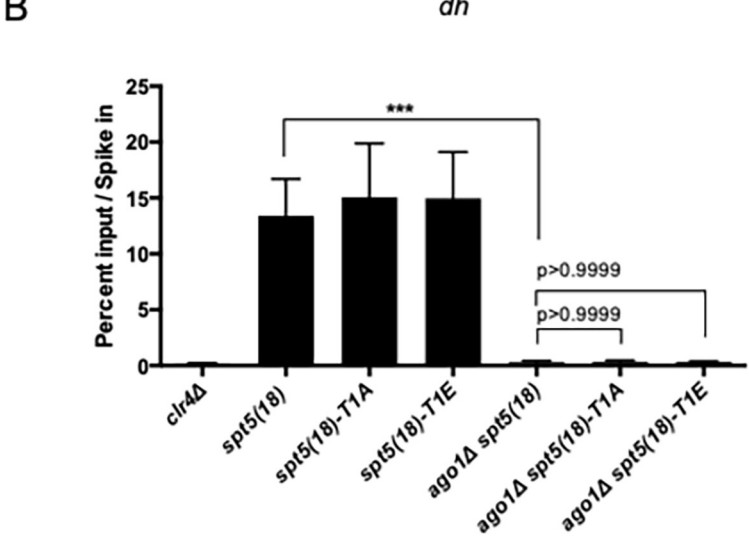

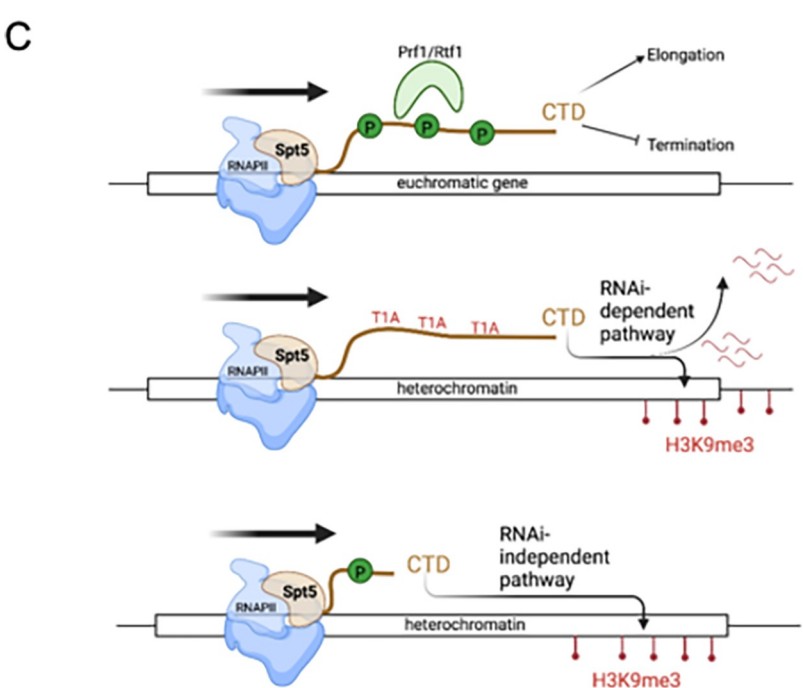

**Fig 4. Spt5 CTD T1 mutations do not rescue heterochromatin formation in the absence of the RNAi pathway. (A)** Spot tests of the indicated strains on control media (YES) or YES containing thiabendazole (TBZ). Plates were incubated for 3 days at 30 degrees before imaging. **(B)** Anti-H3K9me3 ChIP-qPCR quantified with primers amplifying the *dh* repeat in the indicated strains. Percent input was normalized to *S. cerevisiae* spike-in. Error bars indicate SEM. Asterisks indicate significant differences for the indicated comparisons [p<0.05 for 1 star, p<0.01 for 2 stars, p<0.001 for 3 stars, one-way ANOVA followed by Holm-Sidak's multiple comparison test (unpaired) with a single pooled variance; n = 3). **(C)** Cartoon model depicting the roles of Spt5 CTD phosphorylation and Prf1/Rtf1 at a typical euchromatic gene (top), as well as the predicted effects of Spt5 CTD phospho-ablation (middle) or Spt5 CTD truncation (bottom) on heterochromatin regulation.

*ago1Δ prf1-TAP*, reduced to ~15-20-fold in *ago1Δ prf1R227A-TAP* and *ago1Δ prf1R262E-TAP*, and reduced to ~6-fold in *ago1Δ prf1ΔC* (Fig 5C). This partial suppression by *prf1R227A* and *prf1R262E* is similar to what we observed for *spt5-T1* mutants in the context of the truncated CTD (Fig 3). We suspect that the more robust reduction in heterochromatic transcripts caused by *prf1ΔC* is related to a role for the Prf1 C-terminal region in *dh* repeat transcription. Overall, our experiments showed that Plus3 domain mutations and Spt5-T1 mutations had similar effects on heterochromatin, consistent with a role for Prf1 as an effector of pSpt5.

## Synergy between mutations in the Prf1 Plus3 domain and the Spt5 CTD is governed by length of the Spt5 CTD array

Our previous work demonstrated that *prf1-R227A spt5-T1A* or *prf1-R262E spt5-T1A* double mutants exhibited cell separation and drug sensitivity phenotypes that were not observed in

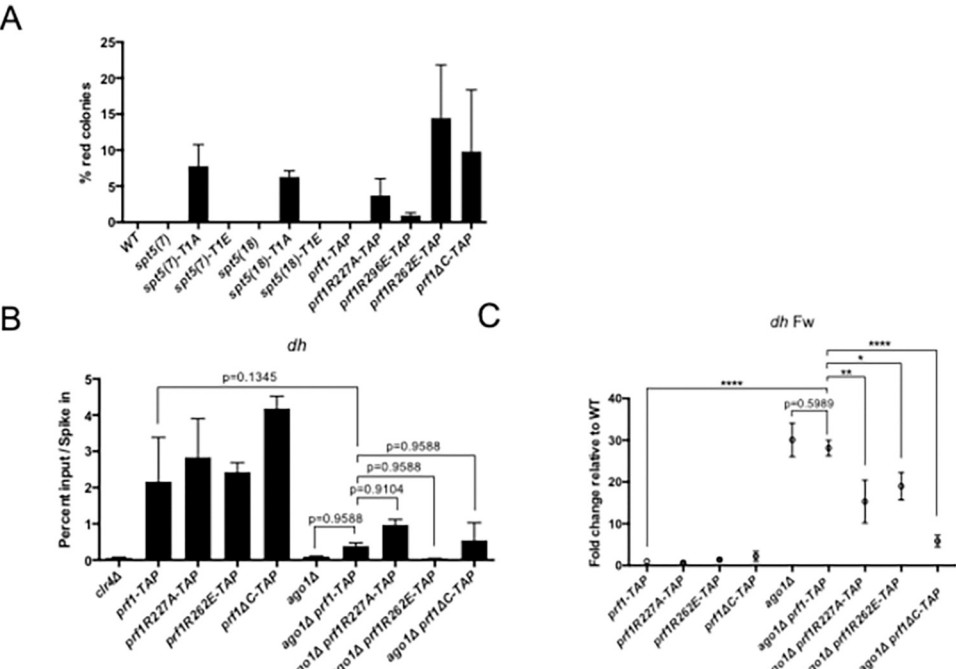

**Fig 5. Evidence that negative effects of pSpt5 on heterochromatin are mediated by Prf1/Rtf1. (A)** Frequency of red colonies formed by the indicated strains expressing *ade6⁺* from the *dg* locus. ~1000 cells from each genotype were plated on adenine-limiting media. Error bars indicate SEM (n = 4). **(B)** Anti-H3K9me3 ChIP-qPCR quantified with primers amplifying the *dh* repeat in the indicated strains. Percent input was normalized to *S. cerevisiae* spike-in. Error bars indicate SEM. Asterisks indicate significant differences for the indicated comparisons [p<0.05 for 1 star, p<0.01 for 2 stars, p<0.001 for 3 stars, one-way ANOVA followed by Holm-Sidak's multiple comparison test (unpaired) with a single pooled variance; n = 3]. **(C)** Strand-specific RT-qPCR analysis of *dh* repeat forward transcripts in the indicated strains. Transcript levels measured with *dh* primers were normalized to *act1⁺* and the wild-type (WT) normalized value was set to 1 [p<0.05 for 1 star, p<0.01 for 2 stars, p<0.001 for 3 stars, one-way ANOVA followed by Holm-Sidak's multiple comparison test (unpaired) with a single pooled variance; n = 3).

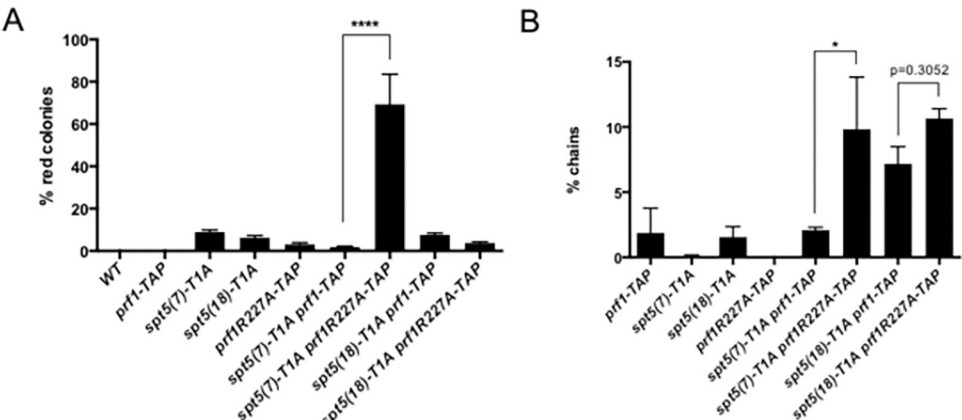

**Fig 6. Synergy between mutations in the Prf1 Plus3 domain and the Spt5 CTD is governed by length of the Spt5 CTD array. (A)** Frequency of red colonies formed by the indicated strains expressing $ade6^+$ from the $dg$ locus. ~1000 cells from each genotype were plated on adenine-limiting media. Error bars indicate SEM (n = 3–4); significance for the indicated comparison (one-way ANOVA) was p<0.0001. **(B)** Frequency of chained cells in log-phase cultures of the indicated strains was quantified by fluorescence microscopy. Each experiment was repeated at least 3 times and >200 cells were counted in each repeat. Error bars indicate SEM; significance for the indicated comparison (one-way ANOVA) was p<0.05.

either single mutant, suggesting that the Prf1 Plus3 domain and pSpt5 had independent functions [68]. These experiments used *spt5-T1A* in the context of the truncated, 7-repeat CTD array. Given the findings presented here, we wished to determine whether a) a similar synergy applied to heterochromatin regulation, and b) the synergy applied to mutant combinations involving *spt5-T1A* in the context of a full-length CTD array. We thus constructed a series of *prf1 spt5* double mutant strains to test these possibilities. We found that the combination *prf1-R227A spt5(7)-T1A* (expressing the 7-repeat CTD) caused a synergistic increase in silencing of euchromatic *ade6⁺* in the dual *ade6⁺* reporter background compared to either mutation alone (~75% red colonies; Fig 6A). In contrast, the *prf1-R227A spt5(18)-T1A* combination (expressing the full-length CTD) was not significantly different from either mutation alone, indicating epistasis (Fig 6A). Thus, both the length and phosphorylation state of the Spt5 CTD repeat array influence Prf1 function in heterochromatin regulation. To determine whether this is the case for Prf1 function more generally, we monitored the frequency of chains of unseparated cells, a cell separation phenotype that is characteristic of *prf1 spt5* double mutants, using fluorescence microscopy [68]. We observed a synergistic increase in the percentage of chains in the *prf1-R227A spt5(7)-T1A* double mutant compared to either single mutant, in agreement with our previous results (Fig 6B). The *prf1-R227A spt5(18)-T1A* double mutant exhibited an elevated level of chained cells comparable to that in the *prf1-R227A spt5(7)-T1A* double mutant (Fig 6B). However, a similar phenotype was observed in a double mutant expressing the wild-type *prf1-TAP* control, indicating that the TAP tag causes some loss of Prf1 function in this assay that is exacerbated by *spt5(18)-T1A* (Fig 6B). Nonetheless, the relationship between *prf1-R227A* and *spt5(18)-T1A* is clearly epistatic in this biological context. These differences between truncated and full-length Spt5 CTD repeat array further support a role for Spt5 CTD repeat number in modulating Prf1 function.

## Discussion

Our results demonstrate that the Spt5 CTD negatively regulates heterochromatin in *S. pombe*. We observed a clear functional separation between different Spt5 alleles with respect to effects

on heterochromatin: substitution of Spt5 T1 in the CTD repeats led to enhancement of heterochromatic silencing through the RNAi pathway, whereas reduction in Spt5 CTD repeat number led to increased RNAi-independent silencing. These findings argue that silencing factors in the RNAi-dependent pathway engage the transcriptional machinery differently from those that are RNAi-independent, and that the biological effects of Spt5 CTD length and phosphorylation can be separated.

To our knowledge, this is the first study to identify separation of function alleles in the same gene that differentially regulate RNAi-dependent and RNAi-independent heterochromatic silencing. Comparison of this unique phenotypic profile to that of other negative regulators of heterochromatin is informative regarding the potential mechanisms involved. Mutations in subunits of the Cleavage and Polyadenylation Factor (CPF) complex, a major mRNA 3'-end processing factor, cause phenotypes similar to those we observed for *spt5-T1A* and *prf1* mutants, as they allow small RNA-dependent trans-silencing but do not enhance (and in some cases impair) RNAi-independent silencing [32,35,39]. These include knockouts of genes encoding the phosphatase module subunits of CPF (*dis2*[+], *ppn1*[+], *swd2.2*[+], *ssu72*[+]) and loss-of-function alleles in genes encoding cleavage module subunits (*yth1*[+], *pfs2*[+], *ctf1*[+]). The CPF mutant phenotypes suggest that slowing of transcript cleavage and release may enable *trans*-silencing by shifting kinetic competition between mRNA 3'-end processing and the RNAi pathway toward the latter [30,32]. The *spt5-T1A* phenotype is not likely due to a decrement in transcript cleavage, since *spt5-T1A* reduces RNAPII processivity and leads to *enhanced* termination [70]. Moreover, removal of Dis2, the Spt5 T1 phosphatase, would be expected to phenocopy a *spt5-T1E* mutant, not *spt5-T1A*. Slow elongation caused by *spt5-T1A* (likely by impairment of Prf1/Rtf1 function) may favor association of the RITS complex with nascent transcripts. Alternatively, *spt5-T1A* may perturb association of other factors with nascent transcripts. This could explain the phenotypes of *trans*-silencing enabling mutants such as those affecting the PAF complex, or the mRNA export factors Mlo3 and Dss1 [30,33]. The cryo-EM-derived structure of a RNAPII elongation complex supports this notion: the Spt5 CTR (bound by the Rtf1 Plus3 domain) and the C-terminal extension of the PAF subunit Leo1 are positioned in close proximity to the RNA exit channel and the exiting RNA on the upstream side of the elongating polymerase [60]. However, *leo1Δ* and *mlo3Δ* enhance both RNAi-dependent and RNAi-independent silencing pathways, indicating that *spt5-T1A* must cause a more subtle or specific structural alteration to the elongation complex [41,71].

The *spt5-T1A* mutation also resulted in aberrant spreading of heterochromatin in *cis*, but only in the context of a truncated, 7-repeat CTD. Thus, traversal of the *irc1L* boundary may involve increased activity of both RNAi-dependent and RNAi-independent silencing activities. Interestingly, the *spt5(18)-T1A* and *spt5ΔC* mutations did not result in aberrant spreading of heterochromatin in cis, despite the fact that they allowed increased 5'FOA resistance in cells harboring a reporter gene outside of the *irc1L* boundary. The basis for repression of the reporter gene in these strains is presently unclear. One possible explanation is that these mutations allow RNAi-dependent establishment of co-transcriptional silencing beyond the *irc1L* boundary, but are not sufficient to trigger aberrant spreading of authentic, H3K9me3-marked heterochromatin [72]. The latter may require additional input from an RNAi-independent pathway.

Truncation of the Spt5 CTD specifically enhanced RNAi-independent silencing. Altered interactions with the nascent transcript could likewise explain this effect, given that RNAi-independent silencing has been linked to premature RNAPII termination. Premature termination is promoted by a suite of termination factors, including the CPF complex, and is opposed by TFIIS, a factor that prevents elongation stalling due to polymerase backtracking [34,35,39,67]. How Spt5 CTD truncation participates in this mechanism remains to be

determined. The fact that T1 mutations in the context of the short CTD weakened bypass of *ago1Δ* for heterochromatin formation suggests that the T1 residue has a dual function: it blocks aberrant spread of heterochromatin linked to RNAi, both in *cis* and in *trans*; and it *promotes* RNAi-independent heterochromatin formation.

Enhanced *trans*-silencing in the *spt5ΔC* variant is consistent with a negative regulatory role for pSpt5: lack of pSpt5 in the absence of the Spt5 CTD likely underlies why *spt5ΔC* and *spt5-T1A* behaved similarly in these assays. We expected that *spt5ΔC* may share phenotypes with the truncated, 7-repeat version of *spt5* as well. However, complete CTD removal did not suppress the heterochromatin defect in *ago1Δ* or allow heterochromatin spreading, despite the fact that CTD truncation did. Germane to this difference is the fact that *spt5ΔC* was slightly TBZ-sensitive on its own, suggesting that CTD truncation beyond a minimum repeat length reveals an additional role in centromere function that masks the inhibitory effect of the CTD on RNAi-independent heterochromatin. In addition, full suppression of *ago1Δ* also required the T1 residue of the CTD repeat, which is absent in *spt5ΔC*.

We found that mutations in *prf1*+ exhibited a similar separation of function as the *spt5-T1A* mutation: RNAi-dependent heterochromatin was enhanced, but RNAi-independent heterochromatin was not. This is consistent with the fact that Prf1/Rtf1 specifically binds to pSpt5. The epistasis we observed in the *prf1-R227A spt5(18)-T1A* double mutant also supports this shared function. However, the synergistic effects we observed in the *prf1-R227A spt5(7)-T1A* double mutant belies a more complex relationship between Prf1/Rtf1 and the Spt5 CTD. Similarly, *prf1* mutations partially suppressed *ago1Δ* effects on heterochromatic transcripts, which we observed for *spt5-T1A* in the context of the truncated CTD. It is possible that CTD repeat number directly affects function of Prf1 even when it is not bound to phospho-T1 through its Plus3 domain (as would be the case in *prf1-R227A spt5-T1A* double mutants). It is noteworthy that Prf1 function is largely intact in the *prf1-R227A* mutant, despite the fact that Prf1 protein is no longer detectable on chromatin by ChIP [68]. Alternatively, the length of the CTD array may influence another phospho-T1 binding factor which functionally overlaps with Prf1.

Whereas Spt5 CTD phosphorylation is known to regulate specific CTD interactions, the impact of repeat number on Spt5 CTD function has not been examined in detail, and likely has important general implications for CTD function in transcription. Based on the properties of the Rpb1 CTD, it is intriguing to speculate that the effects of repeat length are attributable to changes in the formation of transcription foci that form by liquid-liquid phase separation [73]. For the Rpb1 CTD, the number of repeats required to sustain function *in vivo* correlates with the biophysical property of phase separation *in vitro*. Phosphorylation of the repeats alters phase separation behavior, and can have a positive or negative effect on foci formation [74]. Formation of phase separated transcription foci has been primarily implicated in early stages of transcription, and there is little known about how these kinds of structures might regulate post-initiation steps. However, there is evidence that a phase separation mechanism is involved in co-transcriptional histone H2B monoubiquitylation [75]. Further exploring this question, and a potential role for the Spt5 CTD and Rtf1/Prf1, are important future avenues of investigation.

## Materials and methods

### Yeast strains and media

Yeast strains used in this study are listed in S2 Table. To generate the *spt5(18)-T1A* and *spt5 (18)-T1E* strains, in which T1 is replaced by alanine or glutamate in every repeat of the full-length spt5-CTD (JT1047, JT1048, JT1049), integration cassettes were constructed by replacing a *spt5*+ gene fragment (+2398 of the *spt5* ORF to the stop codon) in the pUC19-based

spt5$^{CTD+}$-ura4-spt5$^{3'}$ plasmid [76] with synthetic DNA fragments (Genscript) harboring the relevant mutations. The spt5T1A/T1E –ura4$^+$–spt5$^{3'}$ cassettes were excised and transformed into *S. pombe* using standard methods [77]. Ura+ transformants were selected and analyzed by diagnostic Southern blotting and sequencing of PCR-amplified DNA segments to verify correct integrations.

Single *rtf1* and *spt5* mutant strains harboring *ade6$^+$* reporter genes were constructed by mating and tetrad dissection using standard methods [77]. Presence *of ade6+* within *otr1* in adenine prototroph isolates was confirmed by PCR with primers indicated in S2 Table (Dg'Fw and Ade6'Rv). Double mutant strains harbouring *ade6$^+$* reporter genes were constructed using two strategies. JT1387, JT1390 and JT1391 were constructed using mating and random spore analysis [78]. JT1386 was constructed by transformation of strain JT973 with a *prf1-TAP*::*kanMX6* cassette excised from a plasmid as described previously [68].

The *spt5(7)-WT*, *spt5(7)-T1A*, and *spt5(7)-T1E* alleles were introduced into an *IRC1L*::*ura4$^+$* reporter strain in two steps. First, *per1*::*natMX6* was switched to *per1*::*hphMX6* in strain JT793 (S2 Table) by transformation with a *hphMX6* PCR product amplified from plasmid pFA6-hphMX6 [79] to generate strain JT1030. Next, *natMX6*-marked *spt5(7)-WT*, *spt5(7)-T1A*, and *spt5(7)-T1E* alleles were excised from pUC19-based spt5$^{CTD+}$-nat-spt5$^{3'}$ plasmids (versions of the spt5$^{CTD+}$-ura4-spt5$^{3'}$ plasmids described above which contained a truncated CTD array and a *natMX6* cassette instead of *ura4$^+$*; a gift from B. Schwer) and transformed into strain JT1030. Correct integration was verified by diagnostic PCR and sequencing. The *natMX6*-marked versions of *spt5(18)-T1A* and *spt5(18)-T1E* alleles were generated by transformation of strains JT1047, JT1048, and JT1049 (S2 Table) with a *natMX6* cassette (amplified from pFA6-natMX6 [79] with primers Spt5Ura4'Fw and Spt5Ura4'Rv). These strains (JT1192, JT1193, and JT1194) were crossed to JT1030 to introduce the *IRC1L*::*ura4$^+$* reporter gene.

Double mutants with *ago1Δ* were constructed by mating and random spore disruption as described previously; haploid double mutant isolates were verified by mating tests and PCR [78].

YES and pombe minimal glutamate (PMG) media were as described previously [77]. Thiabendazole (TBZ; Sigma) was used at a concentration of 15 μg/L. 5'-fluoroorotic acid (5'FOA; Sigma) was used at a concentration of 1 g/L in PMG containing 45 mg/L uracil. Spot dilution tests were carried out as described previously [68]. Plates were incubated at 30 degrees for 3 to 7 days before imaging.

## Chromatin immunoprecipitation

Chromatin from $1.5 \times 10^7$ *S. pombe* cells was prepared as described previously [80]. For H3K9me3 ChIP, 50 μL of spike-in chromatin (*S. cerevisiae* BY4743 expressing *TFB1-myc*) was mixed with each 1 mL sample prior to taking a 100 μL input sample. The lysate was precleared by incubating with 15 μL dynabeads M-280 straptavidin (Invitrogen by ThermoFisher), preequilibrated with 1 mL lysis buffer [50 mM Hepes pH 7.5, 150 mM NaCl, 1 mM EDTA, 1% Triton X-100, 0.1% Na deoxycholate, 1 mM PMSF, protease inhibitor tablet (Roche) (one mini tablet per 10 mL)] for 2 hours. For each sample, 20 μL of dynabeads were washed with 100 μL 0.5% BSA in TBS (TBS/BSA) twice. 0.5 μg (2 ul) of H3K9me3 antibody (Diagenode) and 0.5 μL of biotin-myc antibody (Abcam) were diluted into 0.5% BSA in TBS to a total volume of 200 μL, and incubated with beads for 1 hour at 4 degrees. The beads were washed with TBS/BSA containing 5 μM biotin for 10 minutes at 4 degrees twice. The beads were washed twice with 500 μL lysis buffer, and transferred into a new low-protein binding tube. The cleared lysate was incubated with the antibody-bound beads overnight, then beads were washed and eluted as described previously. In the final step of the IP clean up, beads were washed with TE instead of TE + SDS 0.75% in order to reduce the concentration of SDS to 0.4%.

For H3K9me3 ChIP of red and white colonies from *ade6+* reporter strains, a single red or white colony that arose from each strain after growth on PMG low adenine (7 mg/L adenine) plates was grown in 2 mL of PMG low adenine (30 mg/L adenine) overnight. Small cultures were used to inoculate 50 mL cultures in PMG low adenine (30 mg/L adenine) and grown overnight prior to formaldehyde crosslinking and ChIP. No spike-in chromatin was used.

For H3K9me3 ChIP of 5'FOA resistant isolates, a single colony that arose from each strain grown on either PMG or PMG+5FOA (1g/L) was inoculated into small cultures, then 50 ml cultures in the respective media, prior to crosslinking. *S. cerevisiae* spike-in was included as above.

For H3K9me2 ChIP, chromatin was extracted from *S. pombe* cells as described previously [80]. 13 µg of spike-in chromatin (prepared from NIH 3T3 cells) was mixed with each 1 mL sample prior to taking a 100 µL input sample. 3 µg of H3K9me2 antibody (Abcam) was added to the remaining ~900 µL of lysate and the remaining IP steps were carried out as described previously using 20 µL Dynabeads protein G (Invitrogen by ThermoFisher) per sample. In the final step of the IP clean up, beads were washed with TE instead of TE + SDS 0.75% in order to reduce the concentration of SDS to 0.4%.

To reverse crosslinking of ChIP samples, the ~250 µL samples were incubated in a dry bath at 65 degrees overnight. ChIP input samples were diluted with 150 µL TE to a total volume of 250 ul, to reduce SDS concentration to 0.4% before reversing crosslinking at 65 degrees overnight.

Purification of DNA from ChIP samples was as described [80] with the following modifications. Samples were not further diluted with TE following crosslinking, phase extraction steps were carried out using a volume of 250 µL phenol:chloroform:isoamyl alcohol 25:24:1, and then again using the same volume of chloroform. Following phase extraction steps, 3 M sodium acetate (pH 5.3) was added to the aqueous phase to a concentration of 300 µM, and 2 µL glycogen was added to facilitate DNA precipitation. Samples were incubated at -20 degrees for at least 12 hours, then centrifuged at 14000g for 30 min. The DNA was recovered, washed with 1 mL 70% ethanol, dried and resuspended in 50 µL TE, and analyzed by qPCR with the indicated primers (S3 Table).

## Spike-in chromatin preparation

Chromatin from *S. cerevisiae* strain BY4743 expressing *TFB1-myc* was used as a spike-in for H3K9me3 ChIP. The anti-myc antibody (Abcam ab81658) was used to IP *TFB1-myc* and qPCR was performed with primers amplifying the promoter of *S cerevisiae* gene *PMA1* (S3 Table). $3.0 \times 10^7$ *S. cerevisiae* cells were used for chromatin extraction as described previously; 50 µL of *S. cerevisiae* chromatin was added per 1 mL of *S. pombe* chromatin to give a cell:cell ratio of 1:20 [80].

Chromatin from NIH 3T3 cells was used as a spike-in for H3K9me2 ChIP. No additional spike-in antibody was added; the anti-H3K9me2 antibody (Abcam) was used to IP H3K9me2 and detected by qPCR with primers amplifying the promoter of the *M. musculus* gene *Pou5f*. NIH 3T3 cells were grown to confluency in 18 mL DMEM + ITS + P/S in 15 cm plates. Media was replaced with 18 mL 1% formaldehyde in DMEM, and cell cultures were agitated slowly on a shaker for 15 min. 2 mL 1.25 M glycine was added to a concentration of 0.125 mM; plates were agitated slowly for 5 min to quench the crosslinking reaction. Plates were transferred to ice then washed with cold PBS. Cells were transferred by adding 4 mL PBS + 1mM PMSF and centrifuged for 5 min at 800g. The cell pellet was resuspended in 1 mL cell lysis buffer (10mM Tris-HCl pH 8.0, 10mM EDTA, 0.5 mM EGTA, 0.25% Triton X-100 with 1mM PMSF and 5 µL PIC (protease inhibitor cocktail, Sigma P8340) and incubated at 4 degrees for 10 min.

Cells were centrifuged at 800g for 5 min to remove cell lysis buffer and resuspended in 100 μL nuclei lysis buffer, (50mM Tris-HCl pH 8.0, 10 mM EDTA, 1% SDS with 1mM PMSF 5 μL PIC/1mL buffer) then incubated on ice for 15 min. The lysate was sonicated (Bioruptor water-bath sonicator) to shear the chromatin for 15 min, 30 seconds on/off on the high setting. The chromatin was cleared by centrifugation at 14000g for 15 min, then stored at -80 degrees prior to IP. A 10 μL sample was taken for DNA clean up and quantification prior to storage.

### RT-qPCR

Isolation of RNA and strand-specific RT-qPCR were performed as described previously, with 1 μg RNA per strain for *act1*[+] RT and 5 μg RNA for *dh*. [80]. Transcript levels obtained with *dh* repeat primers were normalized to *act1*[+] (S3 Table).

### *Trans*-silencing assays

Control and mutant strains containing the *ade6+* reporter construct were grown in 2 mL YES cultures. Once the cultures reached early log phase ($OD_{600}$ ~0.100–0.300) the cell concentration was determined with a haemocytomer. ~1000 cells per condition were plated on a single PMG low adenine plate (pombe minimal glutamate media with 7 mg/L adenine), and incubated at 30 degrees for 7 days. The plates were stored at 4 degrees for 24 hours prior to imaging. ImageJ software was used to quantify the number of red and white colonies on each plate.

## Supporting information

**S1 Fig. Mitotic stability of the colony color phenotypes for the indicated mutant strains harboring the dual *ade6*[+] reporter system.** For each row of spots, a white or red originator colony from agar plates seeded at single cell density as in Fig 1C was dissociated in 100 ul of water, then 5X serial dilutions were carried out for spot tests on adenine limiting media (PMG with 7mg/L adenine) and grown for 7 days at 30 degrees. An *ade6* mutant (*ade6-M210*) was included for comparison.
(TIFF)

**S2 Fig. Effects of Spt5 CTD mutations on endogenous *ura4*[+] expression and on H3K9me beyond the *irc1R* boundary element. (A)** Spot tests of indicated *spt5* mutants on control PMG media and PMG lacking uracil. Plates were incubated at 30 degrees for 3 days before imaging. **(B)** Anti-H3K9me2 ChIP-qPCR in indicated *spt5* mutants and controls. Primers amplify an intergenic region <1 kb outside of the *irc1R* boundary (osIRC1R). Percent input was normalized to mouse spike-in. *clr4Δ* was included as a negative control. Error bars indicate SEM (n = 3). **(C and D)** As in (B) for anti-H3K9me3 ChIP-qPCR. Percent input was normalized to *S. cerevisiae* spike-in.
(TIFF)

**S3 Fig. Characterization of bypass of the RNAi pathway by Spt5 CTD truncation. (A)** Spot tests of the indicated strains on control media (YES) or YES containing thiabendazole (TBZ). Plates were incubated for 3 days at 30 degrees before imaging. **(B and C)** Anti-H3K9me3 ChIP-qPCR quantified with osIRC1R primers in the indicated strains. Percent input was normalized to *S. cerevisiae* spike-in. Error bars indicate SEM. Asterisks indicate significant differences between indicated comparisons [$p<0.05$ for 1 star, $p<0.01$ for 2 stars, $p<0.001$ for 3 stars, one-way ANOVA followed by Holm-Sidak's multiple comparison test (unpaired) with a single pooled variance; n = 3].
(TIFF)

**S4 Fig. Characterization of silencing beyond the *irc1L* boundary in *prf1* mutants. (A)** Spot tests of indicated *prf1* mutants on control PMG media and PMG with 5'FOA. Plates were incubated at 30 degrees for 7 days before imaging. **(B and C)** Anti-H3K9me3 ChIP-qPCR carried out as in Fig 2 using cells of the indicated genotypes grown in non-selective media (white bars) or 5'FOA-containing media (black bars); *clr4Δ* was included as a negative control. Percent input was normalized to *S. cerevisiae* spike-in. Error bars indicate SEM. Asterisks indicate significant differences for the indicated comparisons [p<0.05, one-way ANOVA followed by Holm-Sidak's multiple comparison test (unpaired) with a single pooled variance; n = 3]. (TIFF)

**S1 Table. Summary of *spt5*⁺ alleles used in this study.**
(DOCX)

**S2 Table. *S. pombe* strains used in this study.**
(DOCX)

**S3 Table. Oligonucleotide primers used in this study.**
(DOCX)

## Acknowledgments

We thank B. Schwer, R. Fisher, E. Bayne, and D. Moazed for providing *S. pombe* strains, and B. Schwer for providing plasmids. We thank B. Schwer, S. Shuman, R. Fisher, P. Parua, and members of the Tanny lab for helpful discussions.

## Author Contributions

**Conceptualization:** Sarah MacKinnon, Jason C. Tanny.

**Funding acquisition:** Jason C. Tanny.

**Investigation:** Sarah MacKinnon, Viviane Pagé, Jennifer J. Chen, Ali Shariat-Panahi.

**Resources:** Ryan D. Martin, Terence E. Hébert.

**Supervision:** Terence E. Hébert, Jason C. Tanny.

**Writing – original draft:** Sarah MacKinnon, Jason C. Tanny.

**Writing – review & editing:** Sarah MacKinnon, Jason C. Tanny.

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
