## [Decision Letter · Decision Letter 0]

20 Jan 2023

Dear Dr Tanny,

Thank you very much for submitting your Research Article entitled 'Spt5 C-terminal repeat domain phosphorylation and length negatively regulate heterochromatin through distinct mechanisms' to PLOS Genetics.

The manuscript was fully evaluated at the editorial level and by independent peer reviewers. The reviewers appreciated the attention to an important problem, but raised some substantial concerns about the current manuscript. Based on the reviews, we will not be able to accept this version of the manuscript, but we would be willing to review a much-revised version. We cannot, of course, promise publication at that time.

Thank you for your patience and my apologies for the long review process. As explained to you earlier by the PLOS Genetics team, we had difficulties finding available experts for a complete review set. Thus, I eventually decided to complement the two existing reviews with my expertise to avoid any further delays.

As you will see, both reviewers acknowledge the significance of understanding how RNA polymerase II and elongating factors control RNAi-dependent heterochromatin formation and find the idea of a separation of function of the Spt5-CDT interesting. However, Reviewer 1 is very critical and raised many points. Besides adding missing references and clarifications in text and figures, several experimental parts need further work. H3K9me levels at the pericentromeric boundaries (Figure 2 and S2B-D) should be re-examined following FOA selection, as the current result is inconclusive. (This seems to be a straight-forward experiment analogous to the red-white selection, as described in Figure 1D). I further share the concern that TBZ growth assays generate indirect results and should be compared with other heterochromatin features, particularly in Figure 5, where prf1 mutants have not been analyzed by H3K9me levels and changes in pericentromeric transcript abundance. Also, H3K9me levels should be examined at other heterochromatin regions, as suggested by Reviewer 2, to assess whether the effects on Spt5-CTD are locus-specific. (Here, I found it puzzling why there was no difference in H3K9me2 levels between WT and clr4 cells at the pericentromeric boundaries, while H3K9me3 showed >10-fold difference for the same locus). Finally, one of the main conclusions of this study is that Prf1 mediates the effects of the CTD on heterochromatin, suggesting that they act in the same pathway. However, I am missing an epistasis analyzing (for example assessing the percentage of ade6-dependent red colony formation). This is critical, as Prf1 seems to affect heterochromatin also in Spt5-CTD independent manner (at least a pericentromeric heterochromatin).

If you decide to revise the manuscript for further consideration at PLOS Genetics, please aim to resubmit within the next 60 days, unless it will take extra time to address the concerns of the reviewers, in which case we would appreciate an expected resubmission date by email to plosgenetics@plos.org.

We are sorry that we cannot be more positive about your manuscript at this stage. Please do not hesitate to contact us if you have any concerns or questions.

Yours sincerely,

Sigurd Braun, PhD

Guest Editor

PLOS Genetics

Wendy Bickmore

Section Editor

PLOS Genetics

Reviewer's Responses to Questions

**Comments to the Authors:**

Reviewer #1: Transcriptionally silent heterochromatin plays important roles in regulation of gene expression as well as maintenance of genome integrity. Studies using a model organism, fission yeast revealed that heterochromatin is formed co-transcriptionally in RNAi-dependent and -independent manners. The mechanism that couples transcription and heterochromatin formation has been reported to include many factors, but details of the function of these factors are still unknow. Here, MacKinnon et al. reported that RNA polymerase II processivity factor Spt5 negatively regulates heterochromatin in S. pombe through its C terminal domain (CTD), which is composed of 18 repeats of an amino acid motif that is phosphorylated by Cdk9. They used two types of Spt5-CTD mutants, substitution of the amino acids that is a target of the phosphorylation (T1A) and truncation of the repeats (7 repeats or complete loss) and analyzed the effects of them on heterochromatin formation. They found that T1A mutation and complete deletion of CTD caused heterochromatin formation at an ectopic site by RNAi, showing that phosphorylation of CTD is important for the prevention of aberrant targeting of RNAi. These mutants also caused the expansion of centromeric heterochromatin over the boundary region. In contrast, truncation of CTD to 7 repeats did not enhance the RNAi-dependent ectopic heterochromatin formation but enhanced RNAi-independent heterochromatin formation at centromeric heterochromatin. They also suggested that a known phosphor-specific Spt5-CTD binding factor Rtf1 mediates the observed function of Spt5 int heterochromatin formation.

The observation of the separation of function (prevention of RNAi-dependent and independent heterochromatin) of Spt5-CTD is interesting as the authors suggested. However, it is easy to expect that Spt5-CTD prevents the aberrant targeting of RNAi machinery to an ectopic site, because the previous results showed loss Spt5-CTD interacting factor Rtf1 caused the aberrant targeting. Moreover, authors only compared the phenotypes of the mutants and made a lot of discussion about the possible mechanisms of Spt5-mediated suppression of heterochromatin formation without concrete results. I think several possibilities raised in the discussion, like involvement of processing of nascent transcripts or transcription elongation speed of RNA polymerase II, should be examined.

The followings are the comments to the manuscript.

1) Page 4, line 1. I think the following papers that describes the involvement of RNA polymerase II in RNAi-dependent heterochromatin formation should be referred.

Kato, Hiroaki, et al. "RNA polymerase II is required for RNAi-dependent heterochromatin assembly." Science 309.5733 (2005): 467-469.

Djupedal, Ingela, et al. "RNA Pol II subunit Rpb7 promotes centromeric transcription and RNAi-directed chromatin silencing." Genes & development 19.19 (2005): 2301-2306.

2) Page 5, line 7. “Prf1/Rtf1” suddenly appeared in the text without explanation what it is. Brief explanation will be helpful for the readers.

3) Figure 1. To discuss the targeting of RNAi, it is important to show the mutations do not affect the amounts of siRNAs produced from ade6 gene inserted at centromeric heterochromatin. Without this information, it is hard to discuss the targeting efficiency.

4) Page 10, lines 3-6. Authors performed H3K9me-ChIP without selection and did not observe increase of H3K9me. To claim that heterochromatin is really expanded and caused the observed FOA resustant phenotype, it is essential to show increase of H3K9me. Why authors did not perform ChIP assay using FOA selection.

5) Fig 2SA-C. Please show the position of primers used in H3K9me-ChIP assay in Fig. 2A.

6) Fig. 5C. TBZ sensitivity is an indirect assay to examine the heterochromatin. The defects of the components of kinetochore also caused TBZ sensitivity. Thus, it is essential to examine the H3K9me level to discuss heterochromatin formation.

Reviewer #2: This manuscript reports a role for phosphorylation of the fission yeast SPT5 C terminal repeat (CTD) in inhibiting de novo RNAi mediated heterochromatin formation. In addition, the authors report a distinct role for truncations of Spt5 CTD repeat number in enhancing RNAi-independent heterochromatin formation. Spt5 is an RNA polymerase II elongation factor. Although previous results had provided evidence for roles of RNA polymerase II and RNA processing or elongation factors in regulation of RNAi mediated heterochromatin, the role of Spt5 in this pathway was unknown. The new results suggest that phosphorylation of threonine 1 of Spt5 CTD negatively regulates RNAi but a truncation of repeat numbers facilitates heterochromatin formation in the absence of RNAi. The authors also show that Spt5 CTD phosphorylation likely acts by recruiting Rft1, providing a mechanistic link between Spt5, Rft1-mediated recruitment of downstream processivity factors, and heterochromatin. The main conclusions of the paper are supported by the experiments the authors present and should be of general interest to the field. The findings begin to define a clearer role for transcription elongation in regulation of heterochromatin-associated events. The paper is suitable for publication in PlosGenetics with minor modifications and clarifications.

Minor points

1. With regards to the role of Spt5 CTD, the authors show that in spt5 CTD-delta ago1-delta double mutant cells the residual H3k9me at the pericentromeric dh repeats is lost (Figure 3B). This suggests that the Spt5 CTD is required for RNAi-independent maintenance of the residual H3K9me. This might be worth mentioning as it points to yet another function of Spt5 in heterochromatin maintenance.

2. Although genome-wide ChIP-seq analysis does not seem essential to support the main conclusions of the paper, it would be interesting to test whether the sub telomeric tlh genes are affected by the various spt5 mutations that the authors have generated.

3. In place of (or in addition to) Figure 4C, it might be more helpful to include a cartoon model that summarizes the various roles of Spt5 in transcription elongation and how these roles relate to different heterochromatin assembly pathways.

**Have all data underlying the figures and results presented in the manuscript been provided?**

Reviewer #1: Yes

Reviewer #2: Yes

PLOS authors have the option to publish the peer review history of their article (what does this mean?). If published, this will include your full peer review and any attached files.

Reviewer #1: No

Reviewer #2: No

---

## [Decision Letter · Decision Letter 1]

6 Oct 2023

Dear Dr Tanny,

Thank you very much for submitting your Research Article entitled 'Spt5 C-terminal repeat domain phosphorylation and length negatively regulate heterochromatin through distinct mechanisms' to PLOS Genetics.

The manuscript was fully evaluated at the editorial level and by independent peer reviewers. In addition to the two previous reviewers, the manuscript was also assessed by a third reviewer. All reviewers appreciated the attention to an important topic and were satisfied with the responses. However, the additional reviewer identified some concerns that we ask you address in a revised manuscript.

In particular, this reviewer raised concerns about the control group(s) for comparing the rescue of the ago1 mutant in the double mutants and noted some discrepancies between statements in the text and figures regarding statistical significance (points 3 and 4). In this regard, I also noted that the use of the comparison group (ago1 or spt5/prf1 single mutant) was not always consistent (e.g., in Figures 3B and 4B, the respective spt5 single mutant was used for stastical comparison; in 3C, the ago1 single mutant was used for the comparison). I agree with the reviewer that the direct comparison between ago1 and the double mutant is more informative than the comparison between the single and double spt5 mutants. One the same line, I would also suggest to consider whether statistical comparisons for WT and spt5 mutants (with and without FOA selection) should be added for Figures 2D and 2E, since the present analysis only describes the additive effect by FOA selection. While this comparison is informative to demonstrate the effect of the selection, it does not allow the comparison with WT cells (which I believe is the focus here). Finally, I also find that a table with all alleles used in this study would be helpful (point 1). The last point (#5) raised by this reviewer is interesting but I consider this as optional.

We therefore ask you to modify the manuscript according to the review recommendations. Your revisions should address the specific points made by each reviewer.

Yours sincerely,

Sigurd Braun, PhD

Guest Editor

PLOS Genetics

Wendy Bickmore

Section Editor

PLOS Genetics

Reviewer's Responses to Questions

**Comments to the Authors:**

Reviewer #1: I think authors adequately answered the points I raised and newly added results support or strengthen their conclusions and hypotheses.

Reviewer #2: The authors have addressed my concerns satisfactorily. I support publication in PlosGenetics.

Reviewer #3: This work examines the involvement of the Spt5-CTD in heterochromatin formation via its regulation of RNAi-dependent and RNAi-independent pathways. The observation that both CTD repeat length and phosphorylation state regulate heterochromatin with separable functions is novel and interesting. The paper does not go in depth regarding the mechanism of this regulation, beyond the expected role of Rtf1. Much work could be dedicated to this mechanism, to try to relate this phenomenon to similar phenotypes found for mutants of the Leo1/PAF complex. However, such efforts would be more informative of the larger role of Spt5/Rtf1 in transcription elongation and termination, which are beyond the scope of this work. As it stands, the experiments presented here set the stage for mechanistic studies that relate the processing of pre-mRNAs as sources and targets of small RNA, and as substrates for the action of RNAi-independent silencing activities. As such, the results presented here have good value that will inform hypotheses going forward. The discussion points to several possible directions for the future. I only have minor comments.

1. The paper makes use of multiple alleles and things get a little confusing at times. The naming scheme is introduced in page 9, and it would be simple enough to introduce each allele individually at that point. In the same vein I found the use of the spt5(18) allele in some figures confusing when used together with WT (Figure 4, S2B). I assume this is the WT allele together with the same marker used to make the other spt5 alleles. Perhaps a table would be useful. Finally, the use of the T1E substitution as a phosphomimetic should be explicitly stated as such; the only explanation of this allele is a later mention of it bearing a constitutive negative charge.

2. Page 15-16: References to Figure 5D should be 5B and 5E should be 5C.

3. In figure 4B the authors show the recovery of H3K9me3 over dh in ago1D mutants by spt5(7) and spt5(7)T1E, showing that the ago1D mutation does not significantly decrease H3K9me3 levels in these mutants. Wouldn’t the correct comparison be ago1D vs ago1D spt5mutant, rather than spt5mutant vs ago1 spt5mutant?

4. In Figure 5B the differences in dh H3K9me3 between prf1-TAP and prf1-TAP ago1D are marked as significant, but the text (page 15) explicitly states them as non-significant. Same for prf1-R227A-TAP ago1D. Since the effect being assayed is the rescue of H3K9me3 in ago1D by the prf1 mutations (see my previous point) this renders this experiment extremely confusing. Correct the figure or the interpretation.

5. The differential effects of these alleles in the forward and reverse dh transcripts is interesting, as it harkens back to observations as old as the original description of RNAi regulation of heterochromatin in Volpe et al 2001, where they showed by run-on assays using swi6D mutants that the forward strand is transcriptionally silenced in WT but the reverse strand is always expressed. The differential regulation of the forward and reverse promoters remains to be elucidated, but Hiten Madhani showed that it could be related to the mechanisms that control nucleosome occupancy at these promoters (Garcia et al G&D 2010). A brief discussion of the potential interaction of the spt6/rtf1 with these mechanisms would be valuable. Could the authors comment on this?

**Have all data underlying the figures and results presented in the manuscript been provided?**

Reviewer #1: Yes

Reviewer #2: Yes

Reviewer #3: Yes

PLOS authors have the option to publish the peer review history of their article (what does this mean?). If published, this will include your full peer review and any attached files.

Reviewer #1: No

Reviewer #2: No

Reviewer #3: No

---

## [Editor Report · Decision Letter 2]

24 Oct 2023

Dear Dr Tanny,

We are pleased to inform you that your manuscript entitled "Spt5 C-terminal repeat domain phosphorylation and length negatively regulate heterochromatin through distinct mechanisms" has been editorially accepted for publication in PLOS Genetics. Congratulations!

Yours sincerely,

Sigurd Braun, PhD

Guest Editor

PLOS Genetics

Wendy Bickmore

Section Editor

PLOS Genetics

Comments from the reviewers (if applicable):

**Data Deposition**

http://datadryad.org/submit?journalID=pgenetics&manu=PGENETICS-D-22-01223R2

**Press Queries**

---

## [Editor Report · Acceptance letter]

2 Nov 2023

PGENETICS-D-22-01223R2 

Spt5 C-terminal repeat domain phosphorylation and length negatively regulate heterochromatin through distinct mechanisms 

Dear Dr Tanny, 

We are pleased to inform you that your manuscript entitled "Spt5 C-terminal repeat domain phosphorylation and length negatively regulate heterochromatin through distinct mechanisms" has been formally accepted for publication in PLOS Genetics! Your manuscript is now with our production department and you will be notified of the publication date in due course.

With kind regards,

Anita Estes

PLOS Genetics

On behalf of:
